# Alismol Purified from the Tuber of *Alisma orientale* Relieves Acute Lung Injury in Mice via Nrf2 Activation

**DOI:** 10.3390/ijms242115573

**Published:** 2023-10-25

**Authors:** Kyun Ha Kim, Soyeon Kim, Min Jung Kwun, Ji Yeon Lee, Sei-Ryang Oh, Jun-Yong Choi, Myungsoo Joo

**Affiliations:** 1School of Korean Medicine, Pusan National University, Yangsan 50612, Republic of Korea; kkha76@hanmail.net (K.H.K.); kmj1205@daum.net (M.J.K.); jy-hj@hanmail.net (J.Y.L.); 2Department of Internal Medicine, Korean Medicine Hospital, Pusan National University, Yangsan 50612, Republic of Korea; omdksy@gmail.com; 3Natural Medicine Research Center, Korea Research Institute of Bioscience and Biotechnology, Cheongju 363-883, Republic of Korea; seiryang@kribb.re.kr

**Keywords:** acute lung injury, alismol, *Alisma orientale* Juzepzuk, mouse model, neutrophilic lung inflammation, NF-κB, Nrf2

## Abstract

Since the ethanol extract of *Alisma orientale* Juzepzuk (EEAO) suppresses lung inflammation by suppressing Nuclear Factor-kappa B (NF-κB) and activating Nuclear Factor Erythroid 2-related Factor 2 (Nrf2), we set out to identify chemicals constituting EEAO that suppress lung inflammation. Here, we provide evidence that among the five most abundant chemical constituents identified by Ultra Performance Liquid Chromatography (UPLC) and Nuclear Magnetic Resonance (NMR), alismol is one of the candidate constituents that suppresses lung inflammation in a lipopolysaccharide (LPS)-induced acute lung injury (ALI) mouse model and protects mice from ALI-like symptoms. Alismol did not induce cytotoxicity or reactive oxygen species (ROS). When administered to the lung of LPS-induced ALI mice (n = 5/group), alismol decreased the level of neutrophils and of the pro-inflammatory molecules, including Tumor Necrosis Factor-alpha (TNF-α), Interleukin-1 beta (IL-1β), Interleukin-6 (IL-6), Monocyte Chemoattractant Protein-1 (MCP-1), Interferon-gamma (IFN-γ), and Cyclooxygenase-2 (COX-2), suggesting an anti-inflammatory activity of alismol. Consistent with these findings, alismol ameliorated the key features of the inflamed lung of ALI, such as high cellularity due to infiltrated inflammatory cells, the development of hyaline membrane structure, and capillary destruction. Unlike EEAO, alismol did not suppress NF-κB activity but rather activated Nrf2. Consequently, alismol induced the expression of prototypic genes regulated by Nrf2, including *Heme Oxygenase-1* (*HO-1*), *NAD(P)H: quinine oxidoreductase-1* (*NQO-1*), and *glutamyl cysteine ligase catalytic units* (GCLC). Alismol activating Nrf2 appears to be associated with a decrease in the ubiquitination of Nrf2, a key suppressive mechanism for Nrf2 activity. Together, our results suggest that alismol is a chemical constituent of EEAO that contributes at least in part to suppressing some of the key features of ALI by activating Nrf2.

## 1. Introduction

Acute lung injury (ALI) is one of the most serious inflammatory lung diseases, featuring severe hypoxemia, diffused neutrophilic infiltration to the lung, alveolar–capillary damage, and a reduction in pulmonary compliance [1]. Although diverse insults can cause ALI, bacterial sepsis is regarded as one of the most common causes of ALI [2,3]. As for treatments, antimicrobial therapy, ventilation, and other supportive care have been known to help improve the survival of ALI patients [4]. However, the mortality remains high, at approximately 40% [5]. One of the main reasons for high mortality is attributable to no effective therapeutics available to treat ALI [4].

The key factor in bacterial sepsis is bacterial endotoxin, lipopolysaccharide (LPS), which is closely associated with the development of ALI [3,5]. The lungs of ALI patients produce various pro-inflammatory cytokines, including tumor necrosis factor-α (TNF-α) and Interleukin-1β (IL-1β) [6], whose production is regulated by LPS binding to TLR4 [7]. Signaling initiated by TLR4 leads to the activation of NF-κB, which results in the production of inflammatory cytokines that exacerbate the symptoms of ALI [7]. Therefore, therapeutic strategies have targeted suppressing TLR4 signaling, NF-κB activity, or the downstream signaling triggered by the inflammatory cytokines [8]. However, these therapeutic strategies have shown limited success [8], suggesting the complexity of inflammation and the necessity of diverse therapeutic approaches. For instance, in addition to NF-κB activation, TLR4 signaling activated by LPS also triggers the production of reactive oxygen species (ROS), which can contribute to tissue damage, furthering inflammation and the symptoms of ALI [9]. As a countermeasure to the deleterious ROS, cells can activate nuclear factor-E2-related factor 2 (Nrf2) [10], a transcription factor that regulates the expression of NAD(P)H: quinine oxidoreductase-1 (NQO-1), heme oxygenase-1 (HO-1), and glutamyl cysteine ligase catalytic units (GCLC) [11], which are involved in removing and guarding against the harmful effects of ROS. Since studies with *Nrf2* knock-out mice have shown a critical role of Nrf2 in protecting mice from ALI [12,13], Nrf2 can be another therapeutic target for treating ALI [14,15].

In traditional Asian medicine, the tuber of *Alisma orientale* Juzepzuk has been prescribed to treat diseases accompanied by edema and inflammation [16]. Accordingly, we have shown that the ethanol extract of *A. orientale* (EEAO) suppresses lung inflammation in an LPS-induced ALI mouse model [17,18]. EEAO can suppress NF-κB activity and concurrently activate Nrf2, both of which were proposed as attributing factors for EEAO to suppress lung inflammation and ALI [18]. In this study, we attempted to find chemical constituents of EEAO that play a role in suppressing lung inflammation. We identified the five most abundant chemical constituents in EEAO by UPLC and NMR analyses and tested whether they have anti-inflammatory activity. Here, we present evidence that among the five abundant chemicals, alismol is one of the chemical constituents of EEAO that plays a role in suppressing neutrophilic lung inflammation manifested in LPS-induced ALI mice. Additionally, we present results indicating that alismol activates Nrf2, which is consistent with EEAO activating Nrf2.

## 2. Results

### 2.1. Alismol Purified from the Ethanol Extract of the Tuber of Alisma orientale Juzepzuk

Since the ethanol extract of the tuber of *Alisma orientale* Juzepzuk (EEAO) was shown to suppress acute lung injury (ALI) in mice [17], we attempted to identify chemicals in the herbal extract that account for the suppressive effect. Fingerprinting analysis of the extract revealed five major peaks, each of which was purified and tested for anti-inflammatory activity. Given that EEAO suppresses NF-κB activity and activates Nrf2 [18], we tested whether the chemicals affect NF-κB or Nrf2 activities by using an Nrf2-dependent luciferase reporter cell line [19] and an NF-kB-dependent luciferase cell line [20], respectively. The preliminary results obtained from the screening experiment suggested that the number 3 peak is a possible anti-inflammatory molecule. Since liquid chromatography–mass spectrometry (LC–MS) and nuclear magnetic resonance (NMR) analyses identified the peak as alismol (Figure 1A), we tested whether alismol could suppress lung inflammation in an LPS-induced ALI mouse model. At the onset of this study, we first determined a cellular toxicity caused by alismol. RAW 264.7 cells were treated with increasing amounts of alismol up to 10^−5^ M, and any cellular toxicity was measured by MTT assay. As shown in Figure 1B, alismol did not show toxicity. In addition, we measured the production of reactive oxygen species (ROS) triggered by alismol, as ROS cause cellular damage, contributing to inflammation [9]. RAW 264.7 cells were treated with vehicle (DMSO 0.01% final), LPS (0.1 μg/mL), or alismol (10 μM final) for 16 h, and intracellular ROS were measured by FACS. As shown in Figure 1C, while LPS strongly induced ROS production, alismol (10 μM) did not trigger ROS production. Together, these results suggest that alismol has no harmful effect on cells.

### 2.2. Alismol Decreases Lung Inflammation in an LPS-Induced ALI Mouse Model

Since ALI is the result of severe inflammatory reactions mostly due to neutrophils [21], we examined whether alismol contributes to the decrease in neutrophil influx to the lung. Since 10 μM of alismol (molecular weight: 220.36) was the maximum amount showing no ROS production and exhibited anti-inflammatory activity in a preliminary screening, we tested whether the equivalent amounts of 10 μM or 1 μM of alismol (2.20 mg/kg mouse body weight: m.b.w. or 0.22 mg/kg m.b.w.) can suppress neutrophilic lung inflammation. As described previously, C57BL/6 mice (n = 5 per group) received an intratracheal (i.t.) injection of PBS or LPS (2 mg/kg m.b.w.), which was sufficient to induce neutrophilic lung inflammation [22]. Two hours after LPS instillation, mice received two different amounts of an i.t. alismol. As shown in Figure 2A, LPS treatment induced the influx of cells into the lung (second column). Differential cell counting revealed that cellular influx to the lung was contributed by mostly neutrophils and macrophages (second column in Figure 2B). However, the cellular influx was tapered down by an i.t. alismol administration (third and fourth columns in Figure 2A and B). Similarly, while an i.t. LPS increased the myeloperoxidase (MPO) activity exerted by neutrophils, an i.t. alismol decreased the MPO activity. Together, these results suggest that alismol suppresses neutrophil influx to the lung caused by an i.t. LPS.

Since neutrophilic infiltration to the lung is correlated with the level of pro-inflammatory cytokines [22], we examined whether alismol decreases the production of inflammatory cytokines that promote inflammation, including TNF-α, IL-1β, and IL-6. From ALI mouse lungs, we collected and analyzed bronchoalveolar lavage fluid (BALF) for detecting excreted cytokines. As shown in Figure 3, LPS induced the production of various cytokines, including TNF-α (the second column in B), IL-6 (the second column in C), MCP-1 (the second column in D), IFN-γ (the second column in E), and IL-10 (the second column in F). However, an i.t. alismol decreased the level of these cytokines in the lung cavity (third and fourth columns in B, C, D, E, and F). To verify that alismol suppresses cytokine production, we harvested mouse lungs, extracted total RNA from them, and analyzed it by quantitative RT-PCR to determine the levels of mRNAs of TNF-α, IL-1β, and IL-6. As shown in Figure 4, while an i.t. LPS increased the mRNA expression of TNF-α (the second column in A), IL-1β (the second column in B), and IL-6 (the second column in C), an i.t. alismol suppressed the mRNA expression of these genes (the third and fourth columns in A, B, and C). As the expression of cyclooxygenase-2 (COX-2) is closely associated with increased inflammation [23,24], we also examined whether alismol suppresses the expression of COX-2. As shown in Figure 4D, while LPS induced the expression of COX-2 (the second column), the two different amounts of alismol suppressed the expression of the gene (third and fourth columns). Taken together, these results suggest that alismol can suppress neutrophilic lung inflammation elicited in the ALI mouse model.

### 2.3. Alismol Protects Mice from ALI-Associated Lung Damage

Major features of ALI include an inflamed lung structure and lung tissue damage, which sometimes result in death [3,25]. Thus, we tested whether alismol relieves these characteristics of ALI. First, we examined whether alismol suppresses the development of LPS-induced inflamed lung structure. Histologic analyses show that, as opposed to PBS-treated control lung (Figure 5A), an i.t. LPS injection induced cellular infiltration to the lung, along with hyaline membrane structure (Figure 5B), which developed during acute lung inflammation [26,27]. However, alismol suppressed the cellular infiltration and the formation of the hyaline membrane structure in ALI mice (Figure 5C,D). Next, we collected BALF from the inflamed mice treated with or without alismol and measured blood albumin, indicative of capillary leakage caused by lung tissue damage [27]. As shown in Figure 5E, unlike PBS-treated mice, the level of blood albumin was increased in the LPS-treated lungs (first and second columns), which was, however, suppressed by alismol treatments (third and fourth columns), suggesting that alismol ameliorates tissue damage was caused by LPS. Taken together, these results suggest that alismol can protect mice from LPS-induced lung damage.

### 2.4. The Anti-Inflammatory Effect of Alismol Is Associated with Nrf2 Activity

Since our results indicate that alismol can suppress inflammation, we set out to explore possible mechanisms that are involved in regulating inflammation. Given that alismol is a constituent of EEAO that activates NF-κB, a central factor regulating inflammation [28,29], we first tested whether the anti-inflammatory activity of alismol was related to suppressing NF-κB activity. RAW 264.7 cells were treated with different amounts of alismol for 16 h and then with LPS (0.1 μg/mL) for 30 min to activate NF-κB. As NF-κB, when activated, moved to the nucleus, nuclear proteins were fractionated, and the nuclear p65 RelA, a subunit of NF-κB [29], was analyzed by Western blotting. As shown in Figure 6A and 6B, alismol did not affect the nuclear localization of NF-κB, suggesting that the anti-inflammatory activity shown by alismol was not related to regulating NF-κB activity. Similarly, we tested whether alismol affected the expression of A20, a cytoplasmic ubiquitin-modulating protein that suppresses multiple inflammatory signaling pathways [30]. RAW 264.7 cells were similarly treated with alismol for 16 h, and cytoplasmic proteins were fractionated and analyzed by Western blotting for A20. As shown in Figure 6C and 6D, alismol did not induce the expression of A20, suggesting that A20 was not the factor that alismol utilized to suppress inflammation. Next, we tested whether alismol suppressed inflammation by activating Nrf2, a transcription factor that contributes to down-regulating inflammatory response [13] and becomes activated by EEAO [18]. RAW 264.7 cells were treated with alismol, as described above. The nuclear fraction was fractionated and analyzed by Western blotting for Nrf2, as Nrf2, when activated, moves to the nucleus [31,32]. As shown in Figure 7A and 7B, as low as 10^−7^ M of alismol could activate Nrf2. To confirm that alismol activated Nrf2, we examined whether alismol suppressed the ubiquitination of Nrf2 to activate Nrf2, given that Nrf2 became inactivated by ubiquitination [15]. HEK 293 cells were transfected with plasmids encoding V5-tagged Nrf2, Flag-tagged Keap1, and HA-tagged ubiquitin. The transfected cells were treated with two different amounts of alismol for 16 h. Two hours before cell harvest, MG132 was added to the cells to preserve ubiquitinated proteins. From a total cell lysate, Nrf2 was precipitated by an anti-V5 antibody, and the status of the ubiquitination of Nrf2 was revealed by Western blotting for HA-ubiquitin. As a ubiquitin assay (Figure 7C) and a densitometric analysis (Figure 7D) show, while the presence of Keap1 heavily ubiquitinated Nrf2 (lane 2), alismol treatment suppressed it (lanes 3 and 4). Concurrent with these findings, quantitative RT-PCR analyses show that alismol induced the expression of HO-1 (Figure 7E), GCLC (Figure 7F), and NQO-1 (Figure 7G), prototypic genes, whose expression is regulated by Nrf2. Therefore, our results indicate that alismol can activate Nrf2 by blocking the ubiquitination of Nrf2, suggesting that alismol activating Nrf2 is, at least in part, accountable for the suppression of lung inflammation by alismol.

## 3. Discussion

Since the tuber of *Alisma orientale* Juzepzuk has been prescribed to relieve inflammatory symptoms [16], and the ethanol extract of *A. orientale* (EEAO) suppressed lung inflammation in an LPS-induced ALI mouse model [17,18], the herb likely contains chemicals as its constituents that exert an anti-inflammatory effect. Thus, in this study, we set out to identify the chemical constituents accountable for this effect. EEAO was subjected to UPLC analysis, from which five major peaks were selected and tested for suppressing NF-κB activity or activating Nrf2, given that EEAO suppresses NF-κB activity as well as activates Nrf2 [18]. We found that alismol purified from EEAO was a constituent that suppressed lung inflammation and protected mice from ALI-associated lung damage.

Our ALI mouse model, in which LPS was administered in aerosol to the lung, could recapitulate some of the key features of ALI. One of them is a diffused neutrophil infiltration to the lung [21]. Since neutrophil is an innate immune cell that is exclusively involved in inflammation, an influx of neutrophils to the lung is indicative of lung inflammation that occurred after LPS administration. Consistent with this notion, LPS administered to the lung also induced the production of prototypic pro-inflammatory cytokines, such as TNF-α, IL-1β, IL-6, MCP-1, and IFN-γ. With this mouse model, we found that a single i.t. alismol lowered the number of neutrophils in the lung cavity and consequently suppressed MPO activity, which mostly pertains to neutrophils. Alismol also decreased the levels of inflammatory cytokines and ameliorated the inflamed lung, which was infested with inflammatory cells and the hyaline membrane. Another key feature of ALI includes alveolar–capillary damage in the lung [26]. In our ALI mouse model, we found that the lungs developed a capillary leakage because serum albumin was detectable in the BALF of the mouse lungs. However, when alismol was administered to the lung, alismol could reduce the level of serum albumin in BALF, suggesting that alismol protected the lung from capillary damage induced by LPS. Together, these results suggest that alismol is a constituent of EEAO that accounts for suppressing lung inflammation and lung damage.

With our ALI mouse model, we showed that alismol could suppress key features of ALI. However, it was not clear how alismol exerts its effect against ALI-like symptoms. Since EEAO was reported to decrease lung inflammation and ALI by suppressing NF-κB, a pro-inflammatory transcription factor, and by activating Nrf2, an anti-inflammatory transcription factor, we first examined whether alismol inhibits NF-κB activity for suppressing lung inflammation and ALI. However, alismol did not suppress NF-κB activity, given our results that alismol could not prevent the nuclear localization of NF-κB, indicative of activated NF-κB [29]. As an alternative mechanism, we explored the possibility that alismol induces the expression of A20, a ubiquitin-modulating protein that blocks multiple inflammatory pathways [33]. However, alismol did not induce the expression of A20 either. These results together suggest that alismol does not modulate NF-κB for its anti-inflammatory activity. Thus, we examined the possibility that alismol activates Nrf2. Our results show that alismol activated Nrf2 by decreasing the ubiquitination and thus blocking the degradation of Nrf2 [31], a key mechanism of suppressing Nrf2 activity. Consistent with this finding, alismol also supported the expression of representative genes regulated by Nrf2, including HO-1, NOQ-1, and GCLC. It is of note that alismol did not activate Nrf2 by triggering ROS production, given that ROS is a potent activator of Nrf2. Together, our results suggest that alismol prosecutes its anti-inflammatory effect by activating Nrf2.

Unlike EEAO, which regulates both NF-κB and Nrf2, alismol appears to activate only Nrf2. Given that EEAO contains numerous chemical constituents [34,35], it is highly likely that other chemical constituents participate in suppressing NF-κB activity. Interestingly, alismol activating Nrf2 seems sufficient to protect mice from ALI. Considering that Nrf2 plays a key role in ameliorating ALI [12,13] and other respiratory diseases, including asthma [36,37], the activation of Nrf2 could be potent enough to quench inflammatory reactions without additional anti-inflammatory measures, such as suppressing NF-kB. However, since we tested only a few pathways that could affect inflammation and ALI, the possibility that alismol also activates other anti-inflammatory or suppresses other pro-inflammatory pathways cannot be excluded. For instance, alismol decreased the production of IFN-γ in ALI mouse lungs, suggesting the possibility that alismol is involved in regulating IFN-γ expression. Since the expression of IFN-γ is regulated by several transcription factors, including ATF-2 [38], AP-1, or CREB/ATF2 [39], alismol may suppress the activities of these transcription factors, resulting in the decreased production of IFN-γ. Alternatively, given that complicated multi-layered mechanisms, including epigenetic regulation, are involved in the transcriptional regulation of IFN-γ [40], it is also possible that alismol interferes with some of the epigenetic regulations required for the production of IFN-γ. Regardless, the precise mechanisms exerted by alismol can be elucidated only if molecular targets of alismol are fully uncovered. The target molecules can be deduced by pharmacologic modeling approaches, in which the molecular structure of alismol is used as a bait for proteins that exhibit a calculated high affinity with a structural match. Nevertheless, we identified alismol as a chemical constituent of EEAO that activated Nrf2 by blocking the ubiquitination. Given the anti-inflammatory function of Nrf2, we propose that alismol is one of the constituting molecules of EEAO, contributing to the suppression of lung inflammation and ALI.

## 4. Materials and Methods

### 4.1. Plant Material and Isolation of Alismol

The tuber of Alismataceae *Alisma orientale* Juzepzuk was purchased from OmniHerb Corporation (Yeongcheon, Republic of Korea), which was authenticated by Professor C.W. Hahn (Korean Medicine Hospital, Pusan National University, Yangsan, Republic of Korea). The voucher specimen (number: pnukh003) is kept in the herbarium of the School of Korean Medicine at Pusan National University. The ethanol extract of *A. orientale* Juzepzuk (EEAO) was prepared with the powdered specimen (200 g) soaked in 1000 mL of 80% (*v*/*v*) ethanol at 60 °C for 8 h. Passed through a 0.2 μm filter, the resultant extract was lyophilized to 36 g of powder (18% yield). HPLC was performed using an ACQUITY UPLC^TM^ system (Waters Corporation, Milford, MA, USA) equipped with a photodiode array detector. Chromatographic separations were performed on a 2.1 mm × 100 mm, 1.7 μm ACQUITY BEH C18 (Waters Corporation) column. The column temperature was maintained at 35 °C, and the mobile phases A and B were water with 0.1% formic acid and acetonitrile with 0.1% formic acid, respectively. Optimized UPLC elution conditions were as follows: 0–1 min, 5% B; 1–10.5 min, 5–28% B; wash to 13.5 min with 100% B; and a 1.5 min recycle time. The flow rate was 0.4 mL/min. EEAO was filtered on membrane filters with a pore size of 0.20 mm (Millipore, Burlington, MA, USA), and the injection volume was 3 μL. Shown in Figure 1A are the five major peaks of EEAO, and LC–MS and NMR analyses identified the third major peak as alismol. Purified alismol was stored in DMSO (100 mM). Thus, a vehicle used for mock treatment was PBS containing 0.01% DMSO.

### 4.2. Reagents and Antibodies

TLR4-specific LPS (*Escherichia coli* O55:B5) was purchased from Alexis Biochemical (San Diego, CA, USA). MG132 was obtained from Merck Millipore (Billerica, MA, USA). Sulforaphane and antibodies for HA and Flag were obtained from Sigma-Aldrich (St. Louis, MO, USA). An anti-v5 antibody was purchased from Thermo Fisher Scientific (Seoul, Republic of Korea). All other antibodies, including those against p65 RelA, Nrf2, A20, β-actin, and lamin A/C, were from Santa Cruz Biotechnology (Santa Cruz, CA, USA)

### 4.3. Cell Culture

RAW 264.7 and HEK 293 cells purchased from American Type Culture Collection, Rockville, MD, USA) were cultured in Dulbecco’s Modified Eagle’s Medium (DMEM), containing L-glutamine (200 mg/L) (Hyclone; Logan, UT, USA) supplemented with 10% (*v*/*v*) heat-inactivated fetal bovine serum (FBS) and 100 U/mL penicillin and 100 μg/mL streptomycin (Invitrogen; Carlsbad, CA, USA). The cells were maintained in a humidified incubator at 37 °C and 5% CO_2_ prior to this experiment.

### 4.4. Measurement of Cytotoxicity

Possible cytotoxicity elicited by alismol was determined by the Vybrant MTT assay kit and the protocol provided by the manufacturer (Thermo Fisher Scientific). Metabolically active cells were calculated against untreated cells. The experiment was performed in triplicate at least three times independently.

### 4.5. Measurement of Intracellular Reactive Oxygen Species (ROS)

RAW 264.7 cells (1 × 10^6^ cells/well) were treated with 100 μM carboxy-H_2_DCFDA (Molecular Probes, Eugene, OR, USA) for 30 min at 37 °C and analyzed by the BD FACS Canto II system (BD Biosciences, San Jose, CA, USA). Data acquired were analyzed by FlowJo software v10.8.1 (Tree Star, San Carlos, CA, USA).

### 4.6. Acute Lung Injury (ALI) Mouse Model and Lung Analyses

C57BL/6 mice purchased from Samtaco Bio Korea, Ltd. (Osan, Republic of Korea) were housed in certified, standard cages and fed with food and water ad libitum before the experiment. All the procedure regarding mice was accorded to the NIH of Korea Guidelines for the Care and Use of Laboratory Animals. Experiments with mice were approved by the Institutional Animal Care and Use Committee of Pusan National University (protocol number: PNU-2022-0061). For inducing ALI, male mice (n = 5/group) aged 8–10 weeks were anesthetized with Zoletil (Virbac, Carros Cedex, France) and then received a single intratracheal (i.t.) LPS or sterile saline (0.02% DMSO in PBS) and 2 h later an i.t. 0.22 mg/kg or 2.2 mg/kg m.b.w. of alismol. LPS, PBS, or alismol was loaded in MicroSprayer^®^ Aerosolizer-Model IA-1C (Penn-Century, Wyndmoor, PA, USA) and delivered in aerosol to the lung via trachea under visual guidance.

For the lung analysis, mice were euthanized by CO_2_ gas at 24 h after LPS treatment, and bilateral bronchoalveolar lavage (BAL) was performed by two consecutive instillations of 1.0 mL of PBS via trachea with a sterile 24-gauge intravascular catheter. Inflammatory cells in BAL fluid (BALF) were counted and scored for macrophages, lymphocytes, or neutrophils by Hemacolor (Merck, Darmstadt, Germany). One hundred cells per microscopic field and 300 cells in total were counted. After being perfused with saline and inflated with fixatives, the lung was embedded in paraffin, cut into 5 μm slices, placed on charged slides, and stained with hematoxylin and eosin (HE) staining method. Three separate HE-stained lung sections in each mouse were evaluated in 200× microscopic magnifications.

### 4.7. Measurement of Pro-Inflammatory Cytokines and Serum Albumin in BALF

After bronchoalveolar lavage with PBS, BALF was centrifuged to make a cell-free BALF. Using the cell-free BALF, inflammatory cytokines were measured by cytometric bead array (CBA) and the manufacturer’s protocol (mouse inflammation kit, BD Biosciences). Similarly, serum albumin was measured by using a mouse albumin ELISA kit and the protocol suggested by the manufacturer (Abcam, Cambridge, UK).

### 4.8. Myeloperoxidase (MPO) Activity

MPO activity in the lung homogenate was measured by the myeloperoxidase fluorometric detection kit and the protocol of the manufacturer (Enzo Life Sciences Inc., New York, NY, USA).

### 4.9. Western Blot Analysis

Total and nuclear proteins were isolated by 0.5% NP-40 lysis buffer and NE-PER nuclear extraction kit, respectively, as instructed by the protocol of the manufacturer (Thermo Fisher Scientific). The amounts of proteins were measured by Bradford (Bio-Rad). Equal amounts of proteins were fractionated by NuPAGE gel (Thermo Fisher Scientific) in MOPS running buffer and then transferred to PVDF membrane (Bio-Rad, Hong Kong, China). After incubation with appropriate antibodies, bands of interest were revealed by chemiluminescence (Thermo Fisher Scientific).

### 4.10. Ubiquitination of Nrf2

HEK 293 cells were transfected with expression vectors of HA-Ub, V5-Nrf2, and FLAG-Keap1. At 24 h after transfection, the cells were treated with 10^−6^ M or 10^−5^ M of alismol for 16 h. Three hours before cell harvest, cells were treated with 10 μM of MG132 for 3 h to preserve ubiquitinated proteins. Total cell lysate was prepared with 0.5% NP-40 lysis buffer. The V5-tagged Nrf2 was precipitated with 1 μg of the anti-V5 antibody and, subsequently, with protein A-sepharose (Thermo Fisher Scientific). The final immune complex was analyzed by Western blotting for HA-tagged ubiquitin to reveal the ubiquitinated Nrf2.

### 4.11. Isolation of Total RNA and Real-Time Quantitative RT-PCR

Total RNA from cells or lung tissue was isolated by using The QIAGEN RNeasy^®^mini kit and the protocol of the manufacturer (Qiagen, Hilden, Germany). One μg of RNA was reverse-transcribed to cDNA by M-MLV reverse transcriptase (Promega, WI, USA). PCR was performed using a SYBR Green premixed Taq reaction mixture with gene-specific primers. The forward and reverse primers for TNF-α were 5′-GGTCTGGGCCAT AGAACTGA-3′ and 5′-CAGCCTCTTCTCATTCCTGC-3′; the primers for IL-1β were 5′-A GGTCAAAGGTTTGGAAGCA-3′ and 5′-TGAAGCAGCTATGGCAACTG-3′; the primers for IL-6 were 5′-TGGTACTCCAGAAGACCAGAGG-3′ and 5′-AACGATGATGCACTTGC AGA-3′; the primers for NQO-1 were 5′-GCAGTGCTTTCCATCACCC-3′ and 5′-TGGAGT GTGCCCAATGCTAT-3′; the primers for HO-1 were 5′-TGAAGGAGGCCACCAAGGAG G-3′ and 5′-AGAGGTCACCCAGGTAGCG GG-3′; the primers for GCLC were 5′-CACTG CCAGAACACAGACCC-3′ and 5′-ATGGTCTGGCTGAGAAGCCT-3′; the primers for COX-2 were 5′-CCCAGAGCTCCTTTTCAACC-3′ and 5′-AATTGGCACATTTCTTCCCC-3′; and the primers for GAPDH were 5′-GGAGCCAAAAGGGTCATCAT-3′ and 5′-GTGA T GGCATGGACTGTGGT-3′. Thermal cycling condition consisted of 95 °C for 10 min, followed by 40 cycles of 95 °C for 10 s, 57 °C for 15 s, and 72 °C for 20 s. Real-time PCR was conducted using a Rotor-Gene Q real-time PCR system (Qiagen) with SYBR Green PCR Master Mix (Enzynomics, Daejeon, Republic of Korea). The threshold cycles (Ct) were used to quantify the mRNA expression of the target genes.

### 4.12. Statistical Analyses

One-way analysis of variance (ANOVA) tests with Tukey’s post hoc test was used to compare among groups (with the assistance of InStat v3, Graphpad Software, Inc., San Diego, CA, USA). *p*-values lower than 0.05 were considered significant. All experiments were performed at least three times independently.

## 5. Conclusions

Given the previous report showing that EEAO, an ethanol extract of the tuber of *Alisma orientale* Juzepzuk, suppresses lung inflammation and ALI in an LPS-induced ALI mouse model by blocking NF-κB and activating Nrf2, we sought out chemical constituents accountable for the therapeutic effect of EEAO. We found that alismol purified from EEAO suppresses lung inflammation and other representative features manifested in an LPS-induced ALI mouse model. Mechanistically, alismol activated Nrf2 and induced the expression of Nrf2-dependent genes but did not suppress NF-κB. Based on these results, we proposed that alismol is one of the key chemical constituents of EEAO, contributing, at least in part, to suppressing lung inflammation and ALI by activating Nrf2.

## Figures and Tables

**Figure 1 ijms-24-15573-f001:**
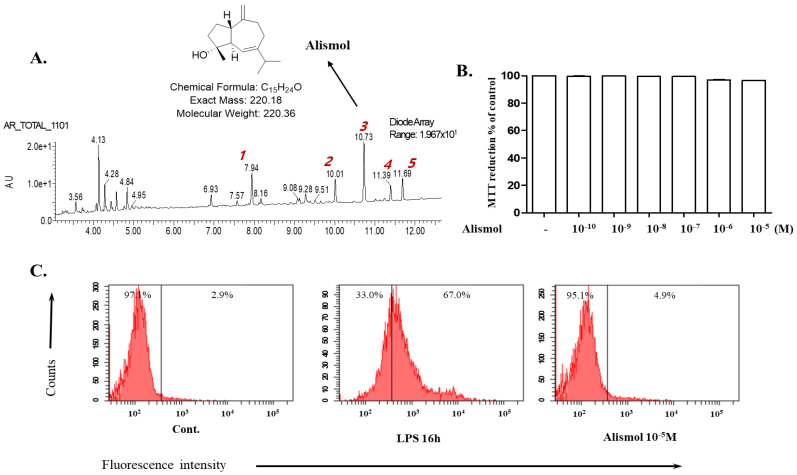
UPLC chromatogram of alismol and its cytotoxicity. (**A**) The peak and the structure of alismol are shown on the UPLC chromatogram of EEAO. Possible harmful effects of alismol on cells were determined by treating RAW 264.7 cells with increasing amounts of alismol up to 10^−5^ M for 16 h and then measuring reduced MTT (**B**) or intracellular ROS (**C**).

**Figure 2 ijms-24-15573-f002:**
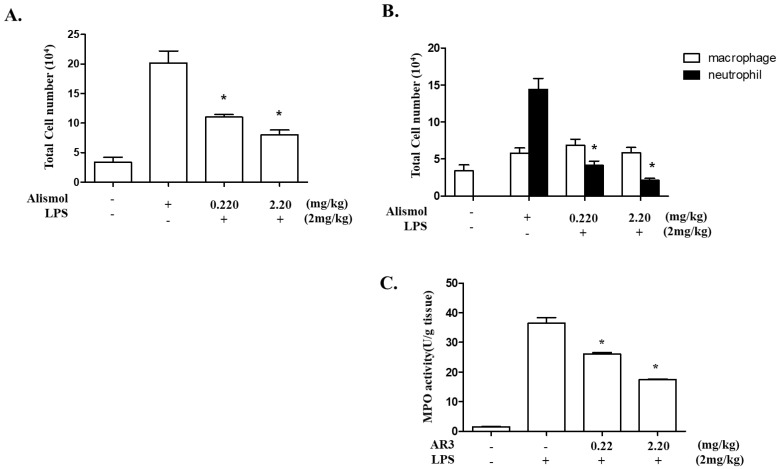
Effect of alismol on neutrophil infiltration to the lung of LPS-induced ALI mice. C57BL/6 mice (n = 5/group) received an i.t. LPS. Two hours later, mice were administered with a single i.t. injection of alismol (0.22 mg/kg or 2.2 mg/kg m.b.w). At 16 h after the i.t. LPS, BALF was collected from the mouse lungs, from which infiltrated cells were harvested and analyzed by differential centrifugation. Total cells (**A**) and macrophages and neutrophils (**B**) in BAL fluid of mice (n = 5/group) were scored. A portion of the right lung was harvested and homogenated, from which MPO activity was measured (**C**). * *p* was less than 0.001 compared with the mice treated with LPS only. Data represent the mean ± SEM of 5 mice.

**Figure 3 ijms-24-15573-f003:**
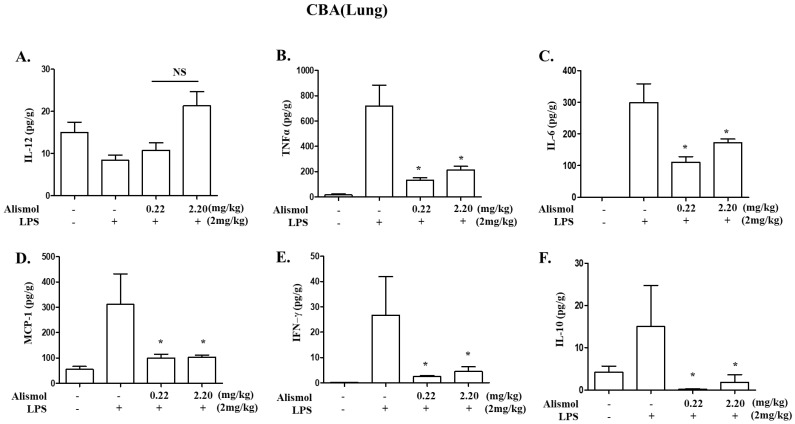
Effect of alismol on the production of inflammatory cytokines in ALI mice. Lung lavage was performed to collect BALF from the lungs of C57BL/6 mice (n = 5/group) that received an i.t. LPS and 2 h later an i.t. alismol (0.22 mg/kg or 2.2 mg/kg m.b.w.). Pro-inflammatory cytokines excreted to the lung cavity were measured by a CBA kit that detects IL-12p70 (**A**), TNF-α (**B**), IL-6 (**C**), MCP-1 (**D**), IFN-γ (**E**), and IL-10 (**F**). * *p* was less than 0.05 compared to that treated with LPS only. Data represent the mean ± SEM of 5 mice.

**Figure 4 ijms-24-15573-f004:**
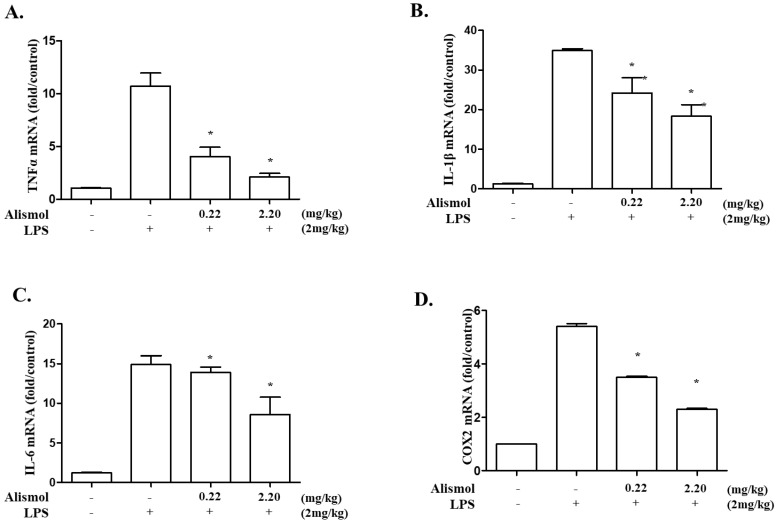
Effect of alismol on the mRNA expression of inflammatory cytokines in ALI mice. From the lung tissue of mice treated with LPS along with alismol, total RNA was extracted and analyzed by quantitative real-time RT-PCR for a relative expression of TNF-α (**A**), IL-1β (**B**), IL-6 (**C**), and COX-2 (**D**) over GAPDH. * *p* was less than 0.05 compared to that treated with LPS only. Data represent the mean ± SEM of 5 mice.

**Figure 5 ijms-24-15573-f005:**
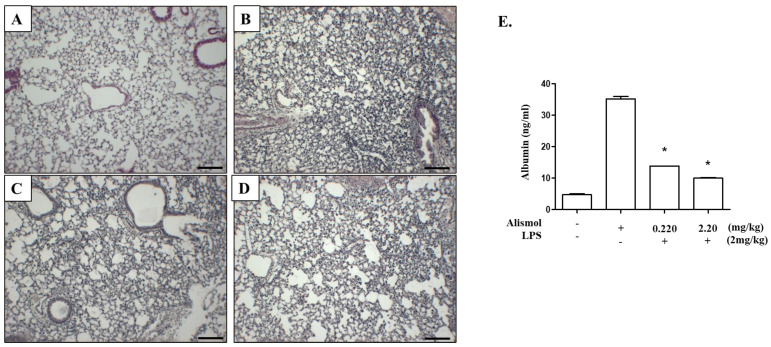
Alismol suppresses the key features of an LPS-induced ALI mouse model. C57BL/6 mice (n = 5/group) received PBS (**A**) or a single, 2 mg/kg m.b.w. of i.t. LPS (**B**–**D**). At 2 h after LPS treatment, mice received a single, 0.22 mg/kg m.b.w. of i.t. alismol (**C**) or 2.2 mg/kg m.b.w. of i.t. alismol (**D**). At 24 h after LPS administration, the lungs of mice were harvested and stained with HE for histological examination. Data are representative of at least five different areas of a lung (bar = 50 μm, 100× magnifications). (**E**) Bronchoalveolar lavage was performed with the mice treated as above. The resultant BALF was centrifuged to obtain cell-free BAL fluid (2 mL), in which albumin was measured by ELISA. * *p* were less than 0.05 compared to that treated with LPS only.

**Figure 6 ijms-24-15573-f006:**
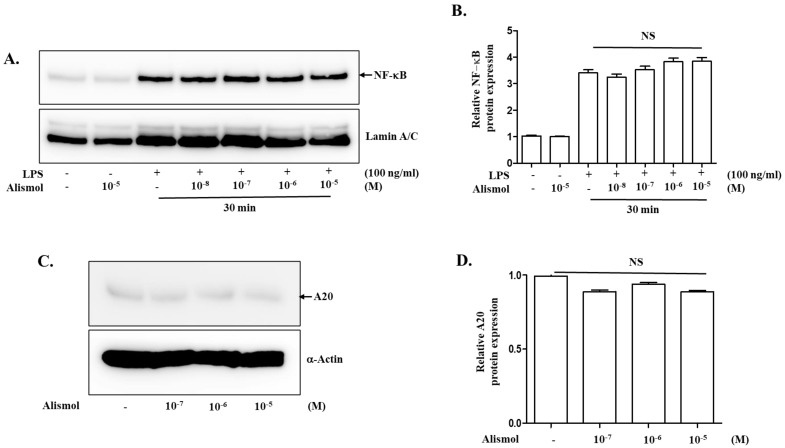
No effect of alismol on NF-κB activity. (**A**) RAW 264.7 cells were treated with different amounts of alismol from 10^−9^ M to 10^−6^ M for 16 h. Before harvest, cells were treated with LPS (100 ng/mL) for 30 min to activate NF-κB. Nuclear proteins were fractionated and analyzed with an anti-p65 RelA antibody to reveal activated NF-κB. The membrane was stripped and reprobed with an anti-lamin A/C antibody to ensure an equal loading of samples. The blots were analyzed by a densitometer, relative amounts of p65 RelA over lamin A/C were measured, and there was no statistical difference among the groups (**B**). (**C**) Similarly, RAW 264.7 cells were treated with different amounts of alismol from 10^−7^ M to 10^−5^ M for 16 h. Cytoplasmic proteins were fractionated and analyzed with an anti-A20 antibody. The membrane was stripped and reprobed with an anti-actin antibody to show equal loading. The blots were analyzed by a densitometer, relative amounts of A20 over α actin were measured, and there were no statistical differences among the groups (**D**). A similar experiment was performed three times in total, and representative results are shown.

**Figure 7 ijms-24-15573-f007:**
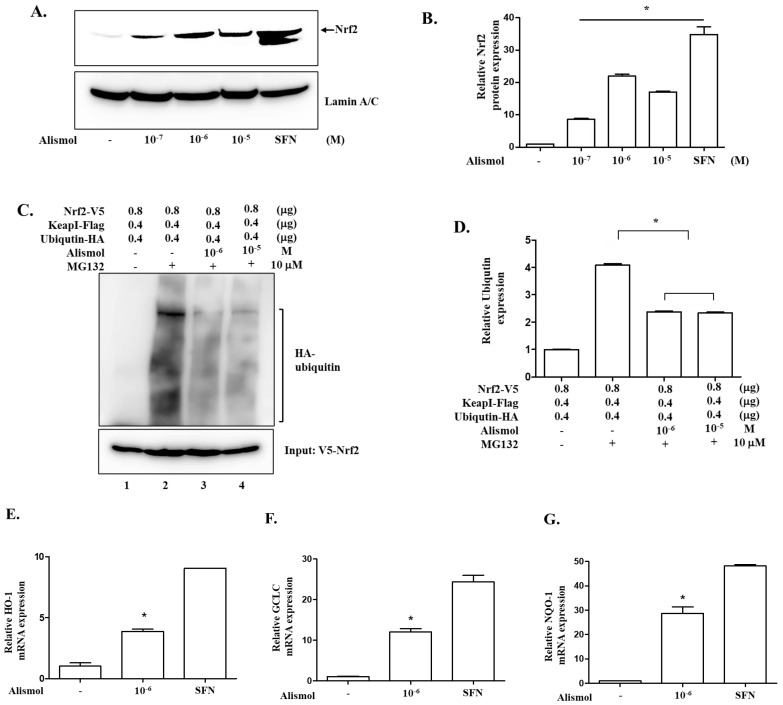
Alismol activates Nrf2 by suppressing the ubiquitination of Nrf2. (**A**) Nuclear proteins were fractionated from RAW 264.7 cells treated with alismol for 16 h and analyzed by Western blotting for Nrf2. As for activated Nrf2, cells treated with sulforaphane (SFN; 5 μM), a potent activator of Nrf2, were included. The membrane was stripped and reprobed with an anti-lamin A/C antibody to ensure an equal loading of samples. Each blot was analyzed by a densitometer, and relative amounts of nuclear Nrf2 over lamin A/C were shown in (**B**). * *p* was less than 0.05 compared to that of the SFN-treated. (**C**) HEK 293 cells were transfected with plasmids encoding V5-tagged Nrf2, Flag-tagged Keap1, and HA-tagged ubiquitin without or with treatment of alismol (10^−6^ M or 10^−5^ M) for 16 h. Three hours before cell harvest, the transfected cells were treated with MG132, a blocker of a ubiquitin-dependent degradation of proteins. V5-tagged Nrf2 was precipitated with an anti-V5 antibody, the complex of which was analyzed by Western blotting for ubiquitinated Nrf2 with an anti-HA antibody. About 10% of total cell lysate was blotted for V5-tagged Nrf2 as an input. Each blot was analyzed by a densitometer, and relative levels of ubiquitination were determined over the input. * *p* was less than 0.05 compared to the untreated (**D**). RAW 264.7 cells were treated with 10^−7^ M of alismol for 16 h. Total RNA was extracted and analyzed by quantitative real-time RT-PCR for HO-1 (**E**), GCLC (**F**), and NQO-1 (**G**). * *p* was less than 0.05 compared to that of PBS-treated. Data represent the mean ± SEM of 3 independent experiments.

## Data Availability

The data presented in this study are available upon request from the corresponding author.

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
