# Peer review of "Alismol Purified from the Tuber of Alisma orientale Relieves Acute Lung Injury in Mice via Nrf2 Activation"

_ijms, 2023, doi:10.3390/ijms242115573_

Round 1

Reviewer 1 Report

The manuscript entitled “Alismol purified from the tuber of Alisma orientale relieves acute lung injury in mice via Nrf2 activation” falls within the scope of the Journal. However, this reviewer has the following comments for the manuscript.

Major comments:

- Authors should insert data related to cytotoxicity in a regular Figure. Did the authors also perform in vivo toxicity studies?

- In Figure 2 authors reported the total cell number. Are the authors sure that the unit of measurement shown on the y-axis is correct (x104)?

- For Figure 2B, do the authors also have data on the number of T lymphocytes?

- Why did the authors use the doses of 0.220 and 2.20 mg/kg for alismol? - The selected doses/concentrations do not meet the standards commonly recognized as necessary for research on the topic of pharmacology. A rationale for the selection of the compound/drug for study as well as for the concentrations/doses employed. Quantities used for concentration- and dose-response experiments should vary logarithmically, e.g., 1, 3, 10, 30 mg/kg, 0.1, 1.0, 10, 100 nanomolar, etc. Justification must be provided for studying only a single concentration or dose of a compound, especially as it relates to reference standards and antagonists/modulators of receptors, enzymes and signalling pathways. Justification must also be provided for the selection of the statistical tests employed as they relate to the experimental design. It is expected that all findings have been subjected to rigorous quantitative analyses, with the calculation and reporting of IC50, Ki, EC50, etc., values. These must be derived from a minimum of three (3) separate and distinct experiments, with the replicates within any single experiment being averaged to obtain a single value for that experimental series. Authors must justify the choice of dose for LPS as well, including the literature reference.

- Did the authors use a positive control to compare the effect of alismol? What is the vehicle used for alismol?

- For Figures 6  and 7 the data are incomprehensible. Authors should represent the results differently and add a densitometric analysis.

- Authors studied the effects of alismol on pro and anti-inflammatory cytokines production. Considering the results, in my opinion, authors should perform a western blot for COX-2 expression.

Minor comments:

- Authors should insert a graphical abstract that summarizes the contents of the article in a concise form in order to capture the attention of the readership

- Authors should report the keywords in alphabetical order.

- Authors should insert an abbreviation section. The words for which is specified an abbreviation should be written in full the first time they are mentioned.

- The English language has to be extensively revised.

- The English language has to be extensively revised.

Author Response

Dear reviewer

Please see the attachment. Thank you for your input and help.

Reviewer 2 Report

Alismol is available many decades ago and do have some effects in different types of disease models. The authors tried to demonstrate its effects in ALI models and potential signal pathways.  However, the animal model and biochemical experiments are massy, did not  clearly demonstrate its effects and signaling pathways.  More reliable experiments are required to make the conclusion. 

• What is the main question addressed by the research? The authors was trying to address the roles and pathways of alismol in LPS-induced lung injury model. However the data are not professionally supporting the conclusion. • Do you consider the topic original or relevant in the field? Does it address a specific gap in the field? Hardly to say • What does it add to the subject area compared with other published material? The data is not reliable so hardly any to add on this field • What specific improvements should the authors consider regarding the methodology? What further controls should be considered? Need to redo most of the experiments professionally • Are the conclusions consistent with the evidence and arguments presented Not really and do they address the main question posed? Not with the data they provide • Are the references appropriate? OK • Please include any additional comments on the tables and figures. The animal model to test survival is not reliable. The data for pathways presented are not professional and hardly to give any conclusion.

need minor revision

Author Response

Dear reviewer

Please see the attachment. Thank you for your help and input.

Round 2

Reviewer 1 Report

I have read the revised version of the manuscript named "Alismol purified from the tuber of Alisma orientale relieves acute lung injury in mice via Nrf2 activation". The authors have made revisions to this article in accordance with the suggestions of reviewers. I think that the manuscript is worth publishing in IJMS.

Author Response

Dear Reviewer

Thanks for your input and time spent on our manuscript. We are happy to address most of the concerns you raised in the previous review. We appreciate your scientific input and valuable critiques.

Sincerely,

Reviewer 2 Report

The revision makes the manuscript better, but did not address some major concerns. 

1. animal model: " a lethal dose of intraperitoneal (i.p.) LPS (10 mg/kg m.b.w.) and D- (+)-galactosamine hydrochloride (500 mg/kg m.b.w.). At 2 h after the injection, mice received a single i.t. injection of 2.2 mg/kg m.b.w. of alismol. Mouse mortality was monitored for 8 days". This is a typical liver failure model, not lung injury model. However, the treatment is to apply alismol into lungs, and explained as anti-lung inflammation.  This will cause a lot confusion. I suggest the authors to change or delete this model from the manuscript.

2. the molecular mechanisms: the presentation is getting better. But to gain a conclusion for Nrf2 pathways, it is necessary to include the Nrf2 inhibitor in some key experiments, to demonstrate the actual roles in alismol anti-inflammation process and lung protection. 

The rest are accepatible

minor correction

Author Response

Dear Reviewer

Thanks for your input and time spent on our manuscript. We are happy to address most of the concerns you raised in the previous review, although our revision falls short of addressing all the concerns. Given the time constraint allocated for the 2nd revision, with due respect, we would like to explain the points you raised in this revision. Our point-by-point responses are below. We hope we address the issues appropriately, which is acceptable to you. Once again, we appreciate your scientific input and valuable critiques.

Sincerely,

  1. animal model: " a lethal dose of intraperitoneal (i.p.) LPS (10 mg/kg m.b.w.) and D- (+)-galactosamine hydrochloride (500 mg/kg m.b.w.). At 2 h after the injection, mice received a single i.t. injection of 2.2 mg/kg m.b.w. of alismol. Mouse mortality was monitored for 8 days". This is a typical liver failure model, not lung injury model. However, the treatment is to apply alismol into lungs, and explained as anti-lung inflammation.  This will cause a lot confusion. I suggest the authors to change or delete this model from the manuscript.

√ Response: Upon your critique, we realized that we failed to provide a scientific ground that explains why the experiment is relevant to ALI. Since other reviewers did not mention the unnecessity of the data set, we decided to provide a better description of why we performed this experiment, rather than eliminating the result. Now, in the new revision, we put a rationale of the experiment with related references. Please check the manuscript of the 2nd revision on page 8, which was highlighted in red. We hope this serves better to clarify the confusion.

  1. the molecular mechanisms: the presentation is getting better. But to gain a conclusion for Nrf2 pathways, it is necessary to include the Nrf2 inhibitor in some key experiments, to demonstrate the actual roles in alismol anti-inflammation process and lung protection. 

√ Response: Although we have been studying Nrf2 function in lung inflammation and other related diseases, somehow, we haven’t weighed in using those chemicals. In the study, we tried to provide at least three different pieces of evidence to link the anti-inflammatory activity of alismol to Nrf2 activation: alismol triggered Nrf2 translocating into the nucleus; alismol induced the mRNA expression of Nrf2-dependent genes; and alismol suppressed the ubiquitination of Nrf2, which is mediated by Keap1, the well-studied inhibitor protein of Nrf2. We agree with the reviewer’s point that chemical inhibitors would consolidate our results. However, given the backlogged supply from manufacturers we experience currently, I am afraid that we could not finish the experiment within the time given to us for resubmission. As suggested by you, we would include the Nrf2 inhibitors for future Nrf2-related investigations. With due respect, in this study, we believe that the above-mentioned three data sets will provide a reasonable case for Nrf2 involvement in the anti-inflammatory function of alismol. 

Round 3

Reviewer 2 Report

How could you make sure the liver failure model cause mouse death is due to ARDS? Your explanation is too reluctant. It is better to remove this animal model from the manuscript and it wont affect your conclusion.

OK

Author Response

Dear reviewer;

Thanks for your intellectual input. We removed the liver failure model in the final version as you suggested. We changed the description, method, figure, figure legend, and references accordingly. The link between liver failure and ARDS may not reflect ALI properly, which I haven't considered. Your comments led me to consider other sets of data that will address new molecular aspects of ALI when studying ALI or ARDS in the future.  I appreciate your comments on our manuscript and study.

Sincerely,